# Robust Video Moment Retrieval with Reflective Knowledge Distillation

## Abstract

With the huge requirement of video content understanding and editing, Video moment retrieval (VMR) is becoming more and more critical, necessitating models that are adept at correlating video contents with textual queries. The effectiveness of prevailing VMR models, however, is often compromised by their reliance on training data biases, which significantly hampers their generalization capabilities when faced with out-of-distribution (OOD) content. This challenge underscores the need for innovative approaches that can adeptly navigate the intricate balance between leveraging in-distribution (ID) data for learning and maintaining robustness against OOD variations. Addressing this critical need, we introduce Reflective Knowledge Distillation (RefKD), a novel and comprehensive training methodology that integrates the dual processes of Introspective Learning and Extrospective Adjustment. This methodology is designed to refine the model's ability to internalize and apply learned correlations in a manner that is both contextually relevant and resilient to bias-induced distortions. By employing a dual-teacher framework, RefKD encapsulates and contrasts the distinct bias perspectives prevalent in VMR datasets, facilitating a dynamic and reflective learning dialogue with the student model. This interaction is meticulously structured to encourage the student model to engage in a deeper introspection of learned biases and to adaptively recalibrate its learning focus in response to evolving content landscapes. Through this reflective learning process, the model develops a more nuanced and comprehensive understanding of content-query correlations, significantly enhancing its performance across both ID and OOD scenarios. Our extensive evaluations, conducted across several standard VMR benchmarks, demonstrate the unparalleled efficacy of RefKD. The methodology not only aligns with the OOD performance benchmarks set by existing debiasing methods but also, in many instances, significantly surpasses their ID performance metrics. By effectively bridging the gap between ID and OOD learning, RefKD sets a new standard for building VMR systems that are not only more adept at understanding and interpreting video content in a variety of contexts but also more equitable and reliable across diverse operational scenarios. This work not only contributes to the advancement of VMR technology but also paves the way for future research in the domain of bias-aware and robust multimedia content analysis.

## 1 Introduction

Video moment retrieval (VMR) aims to pinpoint and retrieve specific moments from a video in response to natural language queries (Anne Hendricks et al., 2017; Hendricks et al., 2018; Liu et al., 2018; Wang et al., 2022; Lu et al., 2019; Yuan et al., 2019; Zhang et al., 2019). This intricate task, hinging on the seamless integration of visual understanding and language processing, is fundamental for various applications, ranging from assistive technologies to intelligent video surveillance. Notably, the prevailing VMR models are trained on richly annotated datasets, aiming to establish robust correlations between visual content and textual descriptions.

Despite the advancements in VMR models, trained on meticulously annotated datasets to capture these correlations, a closer inspection often unveils a significant dependency on dataset-specific biases, especially the temporal location-aware bias (Yang et al., 2021; Hao et al., 2022). These biases, manifesting as predictable patterns in the training data, can lead to superficial associations

between text and video, giving rise to an illusion of understanding rather than genuine content comprehension. Such models, while effective in familiar, in-distribution (ID) settings, encounter difficulties in out-of-distribution (OOD) scenarios where expected biases diverge, questioning their adaptability and efficacy in broader real-world applications.

The endeavor to mitigate these biases has spurred the development of debiasing techniques aimed at broadening the models' generalization abilities. Nonetheless, these strategies frequently hinge on the assumption of a distinct dichotomy between training and testing distributions, an assumption that might inadvertently detract from the models' proficiency within their training environments.

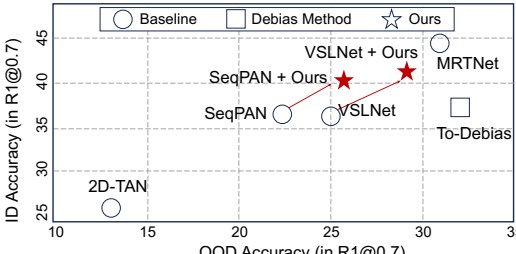

Faced with this challenge, our investigation seeks to address a pivotal question: *How can VMR models be engineered to transcend dataset biases, ensuring robust and reliable performance across both ID and OOD conditions?* Our response is the introduction of Reflective Knowledge Distillation (RefKD), a novel training paradigm that seamlessly integrates Introspective Learning with Extrospective Adjust-

Figure 1: Performance comparison of current video moment retrieval models. Our model, through the novel Reflective Knowledge Distillation (RefKD) approach, enhances the performance of baseline models in both ID and OOD settings, demonstrating the ability to navigate and adapt to varying distribution biases effectively.

ment. This methodology cultivates a dynamic and enriching educational dialogue between the student model and two expert "teacher" models, each reflecting unique bias tendencies from ID and OOD datasets. Through this reflective learning process, the student model is meticulously guided to assimilate and reconcile these diverse insights, achieving a depth of comprehension that surpasses the constraints of dataset-induced biases.

Notwithstanding the potential of RefKD, we acknowledge the challenges posed by potential inaccuracies due to noise in the guidance from teacher models, particularly in OOD contexts. To address this, we have devised a sophisticated label rectification strategy as a core component of RefKD. This strategy involves a dynamic distillation process where the student model is trained not only on direct knowledge from the teachers but also through a co-rectification mechanism that fine-tunes this knowledge. Implementing a side curriculum enables the student model to critically evaluate and refine the received knowledge, bolstering its discriminative capacity and mitigating undue reliance on any single teacher model's perspective.

The contributions of our work in advancing VMR include:

- To the best of our knowledge, we are the first to introduce a paradigm shift in VMR debiasing through Reflective Knowledge Distillation (RefKD), a method that transcends conventional debiasing strategies by advocating for a reflective learning process that encompasses both introspection and extrospection.
- RefKD's dual-teacher framework and its emphasis on introspective learning and extrospective adjustment offer a comprehensive solution to the challenges of bias in VMR, ensuring that models are not just trained but truly educated to understand and interpret content.
- Comprehensive validation of RefKD's efficacy through rigorous experiments on leading VMR datasets, including Charades-CD and ActivityNet-CD. Our results demonstrate RefKD's capability to surpass the performance of existing debiasing methods, setting new standards for VMR system resilience and reliability.

## 2 RELATED WORKS

### 2.1 VIDEO MOMENT RETRIEVAL

Video Moment Retrieval (VMR) focuses on finding video segments that match a given text query, overlapping with other visual retrieval tasks. Fully supervised VMR methods are divided into proposal-based (Anne Hendricks et al., 2017; Hendricks et al., 2018; Liu et al., 2018; Wang et al.,

2022), proposal-free (Lu et al., 2019; Yuan et al., 2019; Zhang et al., 2019), and reinforcement learning-based approaches (Wang et al., 2019) (He et al., 2019).

Proposal-based techniques treat VMR as a ranking challenge, generating segment proposals via sliding windows or networks, then conducting multi-modal semantic matching. However, this results in heavy computational demands. To address this, Yuan et al. (2019) presented a proposal-free approach leveraging a Bi-LSTM and co-attention for multi-modal feature fusion, predicting temporal coordinates directly from sentence queries. Lu et al. (2019) offered a dense bottom-up model, identifying foreground frames linked to queries, then deducing distances to true boundaries, merging suitable temporal candidates. Zhang et al. (2020b) devised a 2D temporal map to grasp temporal relations across moments, signified by start and end timestamps. Reinforcement learning strategies have been applied to fine-tune candidate segments' boundaries to improve query matching. In contrast to existing methodologies, our work explores the synergy between in-distribution (ID) and out-of-distribution (OOD) performance enhancement in VMR, adopting a collaborative approach to address the nuanced challenges inherent in diverse data distributions.

## 2.2 Knowledge Distillation

Knowledge Distillation (KD) is to transfer the knowledge of a teacher model into a student model (Chen et al., 2023; Gou et al., 2021; Niu & Zhang, 2021). The distillation loss forces the features or output of the student approach to those of its teacher. KD has been explored in various cases. To obtain a lightweight model in model compression, KD is to distill a large trained teacher model into a new smaller student model (Hinton et al., 2015). To alleviate the catastrophic forgetting in incremental learning, KD is utilized to transfer the old knowledge of the teacher model to the current student model and the student has the same capacity as the teacher (Masana et al., 2022). To enhance the in-distribution performance of a model in training the model from scratch, self-distillation has attracted a wide interest and the student model itself is directly employed as the teacher model in the next training epoch or stage (Pham et al., 2022). These KD methods have empowered different vision-language tasks, e.g. video question answering (Yang et al., 2020), video captioning (Pan et al., 2020), text-to-image synthesis (Yuan & Peng, 2019). In this paper, we introduces a dual-teacher KD framework, where two distinct teacher models, each attuned to either ID or OOD data biases, guide the student model. This dual-teacher setup fosters a more nuanced learning process, enabling the student model to navigate the intricacies of VMR tasks with enhanced adaptability and performance. By integrating the strengths of both teacher models, our KD framework aims to elevate the student model's performance across a spectrum of data distributions, marking a significant advancement in the application of KD techniques to the VMR domain.

## 3 Method

### 3.1 Preliminary

**Problem Formulation.** Let $\mathbf{V}$ represents an untrimmed video defined as $\mathbf{V} = \{\mathbf{f}_t\}_{t=1}^{T}$, and $\mathbf{Q}$ symbolize a textual query given by $\mathbf{Q} = \{\mathbf{w}_n\}_{n=1}^{M}$. In this context, $T$ and $M$ stand for the count of frames within the video and words in the query, correspondingly. The objective of the VMR task is to pinpoint a segment in the video that aligns with the narrative of the textual query, ascertaining this by determining the beginning and concluding timestamps $(\tau^s, \tau^e)$. Building upon the methodology in previous works (Carreira & Zisserman, 2017), we extract visual features denoted as $\mathbf{V} = \{\mathbf{v}_i\}_{i=0}^{N-1}$ from a dimension of $\mathbb{R}^{n \times d_v}$ utilizing a pre-trained 3D ConvNet (Carreira & Zisserman, 2017). Within this representation, $n$ signifies the count of clips present in the video. In a parallel fashion, query features, represented as $\mathbf{Q} = \{\mathbf{q}_j\}_{j=0}^{M-1}$ from a dimension of $\mathbb{R}^{m \times d_q}$, are derived with the assistance of a pre-trained GloVe (Pennington et al., 2014) embedding. Herein, $m$ stands for the count of tokens or lexemes in the given query.

**Overall Pipeline.** In this work, as depicted in Figure 2, we employ state-of-the-art (SOTA) span-based VMR models (Ji et al., 2022) as our foundational architecture. The model is trained in both conventional and de-biased manners, leading to the emergence of specialized in-distribution (ID) and out-of-distribution (OOD) 'teacher' models. The unique strengths of these teachers are encapsulated in the final logits, which specifically inform the start and end timestamps predictions.

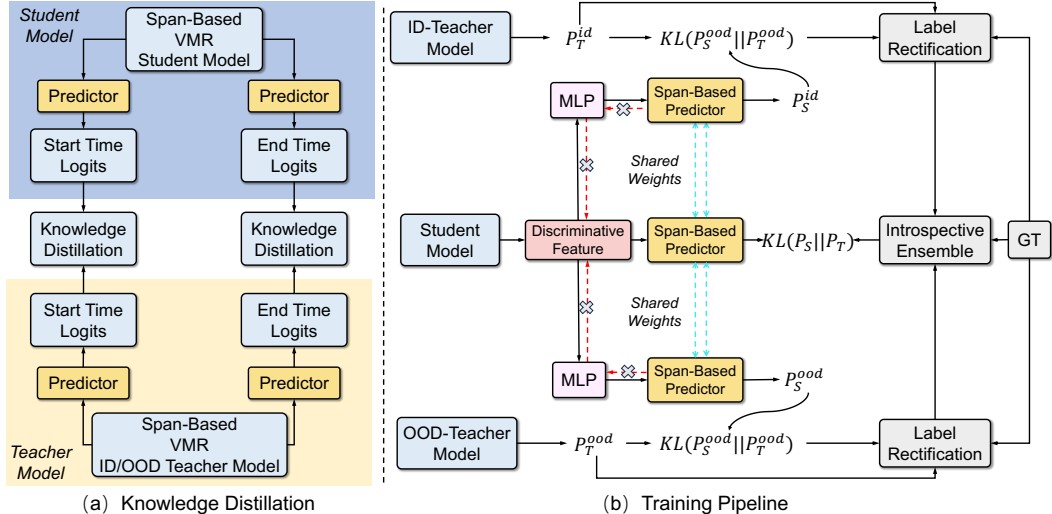

Figure 2: Comprehensive Framework of Our Reflective Knowledge Distillation (RefKD) Approach. (a) Within our RefKD framework, we engage in a concerted effort to distill knowledge from both In-Distribution (ID) and Out-of-Distribution (OOD) teacher models to the student model, focusing on two pivotal aspects: the initiation and conclusion timestamps of video segments. (b) This figure delves into the intricate workings of our RefKD strategy, showcasing how it facilitates a collaborative learning process with insights from both the ID and OOD teacher models. Central to this framework is the Label Rectification Module, which implements a co-rectification mechanism to refine the knowledge transferred from the teacher models, enhancing the student model's learning through self-evaluation. All the training parameters are supervised by Ground Truth (GT).

Firstly, we leverage an "oracle" knowledge derived from an ensemble of both teacher models through introspective integration which is described in Sec. 3.2 during the initial warm-up phase, aiming to provide foundational guidance for the student model to cultivate preliminary discernment. Subsequent to this, using metrics including sample-specific IoU performance, and feedback from a side-curriculum introduced in Sec. 3.3, we refine the soft labels and dynamically modify the composition of the "oracle" knowledge. Ultimately, this intricate and adaptive learning process enables the student model to outperform its teacher counterparts in performance on both the ID and OOD test sets.

**ID-Teacher and OOD-Teacher.** The ID teacher is derived using a standard training paradigm, leveraging unbiased labels and a base model to assimilate knowledge within the distribution. In alignment with Zhang et al. (2021b), the out-of-distribution teacher is procured by employing a training strategy that augments the raw video feature by permuting the location of the ground truth segment, accompanied by two augmented-label consistent loss functions.

## 3.2 INTROSPECTIVE ENSEMBLE

Following Niu & Zhang (2021), we examine the inductive bias through predictions reflecting either ID or OOD inductive biases. Initially, we evaluate the overall confidence associated with ground-truth responses as:

$$c_{s/e}^{\text{ID}} = \frac{1}{\sum_{a \in \mathcal{A}} ||\mathbf{P}_{s/e}^{\text{ID}}(a), \mathbf{GT}_{s/e}^{\text{ID}}(a)||}, \tag{1}$$

$$c_{s/e}^{\text{OOD}} = \frac{1}{\sum_{a \in \mathcal{A}} ||\mathbf{P}_{s/e}^{\text{OOD}}(a), \mathbf{GT}_{s/e}^{\text{OOD}}(a)||}, \tag{2}$$

where indices $s/e$ denote the logits for start or end timestamps, and the operator $||\cdot||$ captures the sample-specific distance between the predicted result within the dataset and the corresponding ground truth label, quantified using the cross-entropy loss.

Subsequently, to incorporate each knowledge aspect, we determine the weights $w_{s/e}^{ID}$ and $w_{s/e}^{OOD}$, premised on the derived confidence scores $c_{s/e}^*$. The underlying rationale posits that the contribution of knowledge to the definitive soft label is inversely proportional to its confidence:

$$w_{s/e}^{ID} = 1 - \frac{c_{s/e}^{ID}}{c_{s/e}^{ID} + c_{s/e}^{OOD}} = \frac{c_{s/e}^{OOD}}{c_{s/e}^{ID} + c_{s/e}^{OOD}}, \tag{3}$$

$$w_{s/e}^{OOD} = 1 - c_{s/e}^{ID} = \frac{c_{s/e}^{ID}}{c_{s/e}^{ID} + c_{s/e}^{OOD}}, \tag{4}$$

Utilizing the derived knowledge weights, the integration of ID-knowledge and OOD-knowledge is articulated as:

$$\mathbf{P}_{s/e}^T = w_{s/e}^{ID} \cdot \mathbf{P}_{s/e}^{ID} + w_{s/e}^{OOD} \cdot \mathbf{P}_{s/e}^{OOD}, \tag{5}$$

Upon synthesizing this amalgamated knowledge sourced from the causal teacher, we then proceed to train a subsequent student model, employing a knowledge distillation approach:

$$\mathcal{L} = KL(\mathbf{P}_{s/e}^T || \mathbf{P}_{s/e}^S) = \sum_{a \in \mathcal{A}} \mathbf{P}_{s/e}^T(a) \log \frac{\mathbf{P}_{s/e}^T(a)}{\mathbf{P}_{s/e}^S(a)}. \tag{6}$$

### 3.3 EXTROSPECTIVE ADJUSTMENT

After completing the Warm-Up phase, the student model has developed a preliminary discriminative capacity and has established its own learning paradigm. It's crucial for this model to critically evaluate the knowledge shared by the teachers, instead of merely absorbing the teacher's foundational knowledge to craft its learning path. To facilitate this, we introduce the concept of side curriculum, which allows the student model to dynamically adjust its learning strategy based on the alignment with both teachers and its own predictive performance.

**Side-curriculum.** The student model includes two extra branched, each tailored to align with a specific teacher.

$$P_{s/e}^{k}{}' = \text{SpanPredictor}(\text{Linear}(\mathbf{f})), \tag{7}$$

where $k \in \{ID, OOD\}$; $f$ is the blended video feature before span prediction; Linear represent the projection from the main branch and $P_{s/e}^{k}{}'$ denotes the side output of the $k$ branch which is designed to fit the side-curriculum.

$$\begin{aligned} \mathcal{L}_{side} =& \lambda_{ID} KL(P_{s/e}^{ID'} || P_{s/e}^{ID}) \\ &+ \lambda_{OOD} KL(P_{s/e}^{OOD'} || P_{s/e}^{OOD}), \end{aligned} \tag{8}$$

$\lambda_{ID}$ and $\lambda_{OOD}$ are hyper-parameters to control the extend of the side-curriculum which are usually small values. The alignment level is employed to determine the confidence scores for each instructor:

$$\begin{aligned} \pi^k =& \frac{1}{\sum_{a \in \mathcal{A}} CE(P_s^{k'}(a), P_s^k(a))} \\ &+ \frac{1}{\sum_{a \in \mathcal{A}} CE(P_e^{k'}(a), P_e^k(a))}, \end{aligned} \tag{9}$$

$$\rho^k = \text{IoU}(P_{s/e}^{k}{}', \mathbf{GT}_e^k), \tag{10}$$

where $\pi^k$ and $\rho^k$ represents the affinity and the confidence of the student model towards the corresponding teacher model. Taking into account the model's affinity and confidence towards the teacher's knowledge, we recalibrate the weighting proportions of the primary curriculum and subsequently refine its content.

**Label Rectification.** In the process of knowledge rectification, we employ the subsequent simple yet effective mixed strategy, incorporated with ground-truth labels, to rectify the imparted knowledge,

particularly in scenarios where the confidence level of a student regarding a particular teacher is suboptimal:

$$\widetilde{\mathbf{P}}^k_{s/e} = \rho^k \mathbf{P}^k_{s/e} + (1 - \rho^k)I_{s/e}, \tag{11}$$

In this equation, $I_{s/e}$ denotes the one-hot label corresponding to the ground truth (GT) labels. The crux of this rectification formula lies in the linear combination of the teacher's labels and the GT labels, modulated by the student's confidence level. This linear composition not only ensures the normalization of the knowledge labels but also refines the labels provided by teachers of lower quality. The underlying principle of reconfiguring labels based on confidence is as follows: if the student model can effectively ground in a given context using the knowledge provided by the teacher, it indicates that the information is beneficial for the student's generalization abilities, regardless of the accuracy of the soft labels provided by the teacher. This approach ensures a balanced integration of teacher-generated labels and ground truth information, optimizing the student model's learning trajectory.

**Dynamic Re-weighting.** To reduce the inductive bias introduced (when one teacher's knowledge nominate the over all "orcale" knowledge, the model will over cultivate the intra partern of the teacher which will lead the student to learn partern of the teacher it self) in the process of the blending two teacher's knowledge. We will re-weight the composition of the "orcale" knowledge. To achieve this goal, we will add more weight to the teahcer which student have lower affinty with it to balance the teacher's correct knowledge in a dynamic way:

$$\begin{aligned}
\widetilde{\mathbf{P}}^T_{s/e} = & \lambda \cdot w^{ID}_{s/e} \cdot \frac{\pi^{OOD}}{\pi^{ID} + \pi^{OOD}} \cdot \widetilde{\mathbf{P}}^{ID}_{s/e} \\
& + \lambda \cdot w^{OOD}_{s/e} \cdot \frac{\pi^{ID}}{\pi^{ID} + \pi^{OOD}} \cdot \widetilde{\mathbf{P}}^{OOD}_{s/e}.
\end{aligned} \tag{12}$$

where $\lambda$ is a normalization parameter to ensure the correctness of the label in numeric scale.

This paradigm reveals that the more affinty with ID teacher the more the knowledge contributed by OOD teacher is and vise versa. The whole training pipeline is summarized in Appendix. 1.

## 4 EXPERIMENTS

### 4.1 DATASET.

Our evaluation is conducted on two benchmark datasets, both of which have been reshaped to account for temporal bias, as described in Yuan et al. (2021).

**Charades-CD.** Derived from the broader Charades dataset (Gao et al., 2017), Charades-CD focuses on temporal sentence localization tasks. Previously split into 12,408 training and 3,720 testing moment-query pairs, we've re-divided it into 11,071 for training, 859 for validation, 823 for test-iid, and 3,375 for test-ood. This restructuring ensures that only test-ood samples deviate from the independent and identical temporal distribution of the training set.

**ActivityNet-CD.** Originating from the comprehensive ActivityNet dataset (Krishna et al., 2017), ActivityNet-CD is specialized for temporal sentence localization tasks. Its rich collection of human activities captured in video format serves as a benchmark for numerous video-centric challenges. To ensure a more balanced and unbiased evaluation, the re-structured annotations contain: 51,415 for training, 3,521 for validation, 3,443 for test-iid, and 13,578 for test-ood. This refined division prioritizes genuine model performance, reducing any advantage from potential temporal biases.

### 4.2 EVALUATION METRICS.

Building upon established methodologies (Zhang et al., 2021b), we employ two key metrics for evaluation: "R$n$@IoU $= m$" and "mIoU." The former measures the top-$n$ retrieved segments that meet or exceed a specified IoU threshold denoted by "$m$". Conversely, "mIoU" calculates the mean IoU across all retrieved segments. For our evaluation on both datasets, we define $n = 1$ and explore $m$ values of 0.3, 0.5, and 0.7.

Table 1: Performance comparison of our proposed RefKD and other SOTA methods on the Charades-CD and Activity-CD datasets.

| Models | | ID Test Set | | | | OOD Test Set | | | |
|---|---|---|---|---|---|---|---|---|---|
| | | R1@0.3 | R1@0.5 | R1@0.7 | mIoU | R1@0.3 | R1@0.5 | R1@0.7 | mIoU |
| Charades-CD | 2D-TAN (Zhang et al., 2020b) | 60.15 | 49.09 | 26.85 | 42.73 | 52.79 | 35.88 | 13.91 | 34.22 |
| | VSLNet (Zhang et al., 2020a) | 72.42 | 55.41 | 35.12 | 50.12 | 63.20 | 43.53 | 24.53 | 42.41 |
| | Shuffle Video (Hao et al., 2022) | 70.72 | 57.59 | 37.79 | 50.93 | 64.95 | 46.67 | 27.08 | 44.30 |
| | To-Debias (Zhang et al., 2021b) | - | 55.66 | 38.87 | 53.92 | - | 50.37 | 32.70 | **50.30** |
| | SeqPAN* (Zhang et al., 2021a) | 67.80 | 57.47 | 37.06 | 49.07 | 61.39 | 44.09 | 23.94 | 41.93 |
| | EMB (Huang et al., 2022) | 71.32 | 59.17 | 38.15 | 51.24 | 63.50 | 48.00 | 28.00 | 44.04 |
| | ID-teacher | 76.79 | 65.37 | 45.20 | 56.35 | 66.81 | 51.85 | 32.47 | 47.14 |
| | OOD-teacher | 74.97 | 62.82 | 44.59 | 56.17 | 68.00 | 52.56 | 33.10 | 47.97 |
| | RefKD (Ours) | **77.64** | **65.49** | **47.51** | **56.92** | **68.98** | **54.58** | **34.49** | 48.92 |
| ActivityNet-CD | 2D-TAN (Zhang et al., 2020b) | 60.56 | 46.59 | 30.55 | 44.99 | 40.13 | 22.01 | 10.34 | 28.31 |
| | VSLNet (Zhang et al., 2020a) | 62.71 | 47.81 | 29.07 | 46.33 | 38.30 | 20.03 | 10.29 | 28.18 |
| | Shuffle Video (Hao et al., 2022) | 63.29 | 48.07 | 32.15 | 47.03 | 42.08 | 24.57 | 13.21 | 30.45 |
| | To-Debias (Zhang et al., 2021b) | - | 40.88 | 28.11 | 44.71 | - | 25.33 | 14.41 | 30.55 |
| | EMB (Huang et al., 2022) | 63.64 | 49.26 | 32.94 | 47.57 | **45.40** | **27.75** | 14.18 | **31.53** |
| | ID-teacher | 64.52 | **49.58** | 33.72 | 48.25 | 39.65 | 23.54 | 12.84 | 28.14 |
| | OOD-teacher | 61.36 | 45.39 | 31.27 | 45.39 | 40.66 | 24.02 | 13.26 | 28.96 |
| | RefKD (Ours) | **64.64** | 49.35 | **33.90** | **48.34** | 42.36 | 25.74 | **14.43** | 30.44 |

### 4.3 IMPLEMENTATION DETAILS.

For our experiments on the Charades-CD and ActivityNet-CD datasets, we select MRTNet (Ji et al., 2022) as our primary model. This model was subjected to two distinct training approaches: standard training and de-biased training, as outlined in Zhang et al. (2021b). To maintain a fair benchmark against prior works (Zhang et al., 2020a;b), we remain consistent in our choice of feature extraction backbones. Specifically, we utilize the I3D (Carreira & Zisserman, 2017) model to extract video features, which was pre-trained and applied without any additional fine-tuning, and the text feature is extracted by pre-trained GloVe (Pennington et al., 2014) embedding. We uniformly sampled 128 clips from each video. We employ the AdamW (Loshchilov & Hutter, 2018) optimizer for training our model, excluding weight decay. Across all datasets, the learning rate was initially set at 0.001 and adjusted according to a linear schedule, which gradually reduced the rate over the course of 100 epochs. and we use a consistent batch size of 32 for both Charades-CD and ActivityNet-CD datasets. In every configuration, the feature hidden dimension is designated as 128.

### 4.4 COMPARISON WITH STATE-OF-THE-ARTS

**Quantitative Results.** To demonstrate the superiority of our proposed technique, we compare it with several state-of-the-art Video Moment Retrieval (VMR) methods under identical experimental settings. These include: 2D-TAN (Zhang et al., 2020b), which introduces a 2D temporal map to model temporal relations between proposals; VSLNet (Zhang et al., 2020a), which uses a context-query attention mechanism for fine-grained multi-modal interaction; Shuffle Video (Hao et al., 2022), which addresses temporal bias by using shuffled videos to improve visual-text alignment and introduces auxiliary tasks for better temporal understanding; EMB (Huang et al., 2022), which uses a Guided Attention mechanism to improve the localization of ambiguous activity boundaries in unscripted video content; To-Debias (Zhang et al., 2021b), which tackles distribution bias by introducing data and model debiasing strategies to improve cross-modal interactions; and SeqPAN (Zhang et al., 2021a), which applies concepts from named entity recognition by segmenting snippet sequences into different regions to improve moment localization.

For fair comparison, we reproduced SeqPAN in a PyTorch environment, consistent with other models, with MRTNet as our main model. The results in Table 1 show that our method not only matches but significantly surpasses state-of-the-art benchmarks across all metrics in both ID and OOD splits, demonstrating its robustness. In the Charades-CD and ActivityNet-CD datasets, RefKD outperformed all models, consistently achieving higher scores in key metrics like R1@0.7 and mIoU. This confirms RefKD's superiority, attributed to its novel techniques that enhance adaptability and accuracy in diverse video moment retrieval scenarios.

| Backbones | Role | Charades-CD (ID) | | | | Charades-CD (OOD) | | | |
| --- | --- | --- | --- | --- | --- | --- | --- | --- | --- |
| | | R1@0.3 | R1@0.5 | R1@0.7 | mIoU | R1@0.3 | R1@0.5 | R1@0.7 | mIoU |
| VSLNet (Zhang et al., 2020a) | ID Teacher | 72.42 | 55.41 | 35.12 | 50.12 | 63.20 | 43.53 | 24.53 | 42.41 |
| | OOD Teacher | 72.54 | 54.92 | 34.39 | 50.92 | 66.46 | 46.93 | 26.64 | 45.29 |
| | **RefKD (Ours)** | **74.24**$_{+2.51\%}$ | **61.97**$_{+11.84\%}$ | **40.95**$_{+16.6\%}$ | **54.06**$_{+7.86\%}$ | **65.51**$_{-1.43\%}$ | **47.53**$_{+1.28\%}$ | **28.47**$_{+6.87\%}$ | **45.49**$_{+0.44\%}$ |
| SeqPAN*(*Zhang et al., 2021a*) | ID Teacher | 67.80 | 57.47 | 37.06 | 49.07 | 61.39 | 44.09 | 23.94 | 41.93 |
| | OOD Teacher | 71.69 | 56.87 | 34.39 | 50.59 | 63.41 | 47.35 | 25.39 | 43.70 |
| | **RefKD (Ours)** | **72.17**$_{+6.45\%}$ | **59.42**$_{+3.39\%}$ | **39.73**$_{+7.20\%}$ | **52.44**$_{+6.87\%}$ | **64.59**$_{+1.86\%}$ | **46.52**$_{-1.75\%}$ | **26.04**$_{+2.56\%}$ | **43.93**$_{+0.53\%}$ |

Table 2: Performance comparison of our RefKD method with different backbone networks. Here * represents the reproduced result of SeqPAN in Pytorch environment.

## 4.5 ABLATION STUDIES

**The effectiveness of proposed distillation strategies.** As shown in Table 3, the ablation studies conducted on the Charades-CD dataset provided insightful findings regarding the efficacy of different distillation strategies in improving model performance. RefKD achieved the best results, particularly in OOD scenarios, outperforming other variants across all metrics, highlighting its robustness. The variant without Label Rectification (w/o LR) showed a slight performance drop, though it had a higher mIoU in ID settings, indicating partial compensation in specific metrics. Removing Dynamic Re-weighting (w/o DR) caused a minor decline across most metrics, emphasizing its role in accuracy. The largest performance drop occurred in the variant without both Label Rectification and Dynamic Re-weighting (w/o both), especially in OOD settings, underscoring the importance of both strategies for adaptability and accuracy.

Table 3: Ablation studies for proposed distillation strategies on the Charades-CD dataset."IntroD" represents basic introspective distillation, "LR" represents Label rectification, and "DR" represents Dynamic Re-weighting.

| Methods | ID | | | OOD | | |
| --- | --- | --- | --- | --- | --- | --- |
| | R1@0.5 | R1@0.7 | mIoU | R1@0.5 | R1@0.7 | mIoU |
| RefKD | **65.49** | **47.51** | 56.92 | **54.58** | **34.49** | **48.92** |
| w/o LR | 64.64 | 47.02 | **57.63** | 54.10 | 33.48 | 48.43 |
| w/o DR | 64.98 | 47.39 | 57.13 | 53.90 | 33.39 | 48.53 |
| w/o both | 64.28 | 46.90 | 55.99 | 53.30 | 33.33 | 48.15 |

**Ablation Studies on Label Rectification Strategies.** In the ablation studies focusing on Label Rectification strategies as detailed in Table 4, our proposed RefKD method, which employs a mixed label strategy, emerges as superior in terms of the R1@0.7 metric across both ID and OOD dataset splits. This underscores the effectiveness of the RefKD method in handling complex scenarios, attributing to its balanced approach in label rectification. The results reveal that the Pure GT (Ground Truth) strategy, which simplifies labels to one-hot GT labels, adversely affects the model's learning process. This approach disrupts the continuity of knowledge representation, leading to suboptimal performance across all metrics, as evident from its lower scores compared to Re-

Table 4: Ablation studies for label rectification method on the Charades-CD dataset. "Pure GT" represents we rectify our labels with GT labels if the IoU of the teacher's prediction is lower than a threshold $\delta = 0.3$, "Window $f$" represents exponential with function.

| Methods | ID | | | OOD | | |
| --- | --- | --- | --- | --- | --- | --- |
| | R1@0.5 | R1@0.7 | mIoU | R1@0.5 | R1@0.7 | mIoU |
| RefKD | 65.49 | **47.51** | 56.92 | **54.58** | **34.49** | 48.92 |
| Pure GT | 64.12 | 45.44 | 55.55 | 52.18 | 32.50 | 48.14 |
| Window $f$ | **65.61** | 46.05 | **56.98** | 53.51 | 33.27 | **49.00** |

fKD. This reinforces the notion that discrete labels without the integration of continuous knowledge can impair the model's holistic understanding and adaptability. Furthermore, the Window function strategy ($f$), while yielding the highest R1@0.5 and mIoU scores in the ID setting, does not maintain this lead in the R1@0.7 metric. This indicates a limitation in its ability to distinguish between closely related labels, especially when the probability of the correct label is low. The strategy's tendency to stretch the GT label while suppressing others can lead to misclassifications, particularly in scenarios where adjacent labels have higher probabilities. Despite these shortcomings, the Window function strategy does offer a marginal improvement in mIoU in the OOD setting, suggesting its potential utility in specific contexts.

**Ablation Studies on Different Backbones.** Our ablation studies on various backbones, as shown in Table 2, demonstrate the versatility and effectiveness of our method across different span-based Video Moment Retrieval (VMR) models. We tested models like VSLNet and SeqPAN, with asterisks indicating PyTorch-reproduced results. Integrating our method with RefKD in VSLNet led to significant performance gains across all metrics, especially in the R1@0.7 metric for the OOD setting of the Charades-CD dataset, with a 3.45% improvement. Similarly, applying our method to SeqPAN also showed notable gains, particularly in the OOD setting. These results highlight our method's robustness, adaptability, and potential to enhance VMR model performance, especially in challenging OOD scenarios.

**Effectiveness of Label Rectification on Noisy Labels.** Here we delve into the critical role of label rectification in enhancing student model performance, particularly when dealing with teacher models exhibiting unstable predictions.

Table 1 indicates that both ID-teacher and OD-teacher show suboptimal performance on the ActivityNet-CD dataset. This observation raises a concern: if the student model directly imitates these unstable teacher models, the excessive noise in their predictions could hinder the student model from acquiring useful knowledge. To investigate this, we analyzed model variant

Table 5: In-depth analysis of Label Rectification Impact on the ActivityNet-CD Dataset.

| Methods | ID | | | OOD | | |
|---|---|---|---|---|---|---|
| | R1@0.5 | R1@0.7 | mIoU | R1@0.5 | R1@0.7 | mIoU |
| Teacher | **49.58** | 33.72 | 48.25 | 24.02 | 13.26 | 28.96 |
| RefKD | 49.35 | **33.90** | **48.34** | **25.74** | **14.43** | **30.44** |
| w/o LR | 47.56 | 32.56 | 46.87 | 22.98 | 12.33 | 27.46 |

without Label Rectification. Results illustrated in third row of Table 5, reveal a significant degradation in model performance if without Label Rectification. To conclude this, Label Rectification enables the student model to dynamically adjust its learning targets, ensuring a more robust and effective distillation process. By mitigating the impact of unreliable teacher model predictions, it plays a vital role in maintaining the integrity and efficacy of the knowledge transfer.

## 5 IN-DEPTH ANALYSIS

Here, we delve into a series of critical inquiries aimed at elucidating the nuanced dynamics and efficacy of our proposed Reflective Knowledge Distillation (RefKD) framework within the context of Video Moment Retrieval (VMR). Through this rigorous analysis, we aim to shed light on the underlying mechanisms and strategic advantages that RefKD introduces to the VMR domain, thereby offering a deeper understanding of its operational principles and potential impacts.

**Q1:Why is OOD an important problem in VMR task?** A1: The challenge of Out-of-Distribution (OOD) in Video Moment Retrieval (VMR) primarily arises from the variability and unpredictability of real-world video content that may not be represented in the training dataset. In VMR, an "In-Distribution" (ID) scenario refers to instances where the model encounters data similar to what it was trained on, including similar contexts, objects, actions, and interactions. Conversely, OOD scenarios involve data that significantly deviate from the training set, which could include novel contexts, unseen actions, or different interactions between objects and subjects within the video. The distribution shift here refers to the change in data characteristics from the ID to OOD contexts, posing significant challenges for VMR models in maintaining high retrieval accuracy.

**Q2: How to have an "OOD teacher"?** A2: The notion of an "OOD teacher" in the context of Reflective Knowledge Distillation (RefKD) is a conceptual tool designed to equip the student model with the ability to generalize beyond the ID data. It's important to clarify that the "OOD teacher" doesn't directly encounter true OOD data during its training. Instead, it is trained in a manner that simulates OOD conditions, often by diversifying the training data, introducing data augmentation techniques, or employing adversarial training methods to expose the model to a wider range of data variations. This training approach enables the "OOD teacher" to develop strategies and insights that are effective in OOD scenarios, which it can then impart to the student model. The goal is to enhance the student model's robustness and adaptability, allowing it to perform more effectively when faced with genuine OOD data in real-world applications.

**Q3: Analysis of Label Rectification.** A3: We delve into an in-depth analysis of the window function label rectification method. Previously, we explored three distinct label rectification strategies: the mixed approach (our proposed method), the pure ground truth (GT) method, and the window function strategy. The window function, denoted as $f$, is mathematically represented as:

$$f(\mathbf{x})_{s/e}^k = e^{-(1-\rho^k)\cdot(\mathbf{x}-I_{s/e})^2\cdot(\tau)^{-1}} \tag{13}$$

In this equation, $\tau$ denotes the temperature parameter controlling the sharpness of the window function. The label rectification process utilizing this function is calculated by:

$$\widetilde{\mathbf{P}}_{s/e}^k = \kappa \cdot \mathbf{P}_{s/e}^k \odot f(\mathbf{P}_{s/e}^k)_{s/e}^k, \tag{14}$$

where $\kappa$ serves as a normalization factor, ensuring the integrity of label logits.

While this strategy effectively rectifies erroneous knowledge, it poses challenges in terms of control and precision. Specifically, the process of enhancing labels corresponding to ground truth inadvertently suppresses other potentially beneficial knowledge imparted by teacher models. This suppression is not merely a side effect; it has substantive implications for the learning process. Specifically, it can detrimentally impact the generalization abilities of the student model. This is particularly evident in our experimental results, where we observed a marked decline in performance on the R1@0.7 metric. This outcome suggests that the indiscriminate suppression of teacher model knowledge while rectifying errors, may also inadvertently filter out valuable generalizable insights.

**Q4: Exploration of Warm-Up Epochs.**

A4: In the structured methodology of our training pipeline, the initial warm-up phase is pivotal, setting the stage for the student model's effective engagement with the knowledge distillation process. This preparatory stage is meticulously crafted to endow the student model with the foundational skills necessary for a critical examination and integration of diverse knowledge sources. As shown in Figure 3, an insufficient or absent warm-up phase significantly harms model performance, with declines of up to 7.52% in R1@0.7 and 12.24% in mIoU. This is due to inconsistent knowledge transfer and weak foundational understanding. A well-calibrated warm-up period over 20 epochs stabilizes performance by building the student model's confidence, enhancing training effectiveness. However, an overly long warm-up can lead to overfitting and a performance plateau, emphasizing the need to balance the warm-up duration for optimal results.

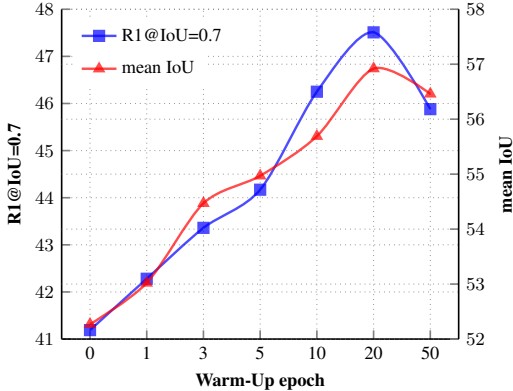

Figure 3: Ablation Studies on the Charades-CD Dataset: Evaluating the Impact of Various Warm-Up Epoch Epochs in MRTNet Before Student Knowledge Re-adjustment from Teachers.

## 6 CONCLUSION

In this work, we presented Reflective Knowledge Distillation (RefKD) as a strategic advancement to fortify the robustness of video moment retrieval (VMR) systems against the prevalent challenges of dataset biases and generalization. Through the innovative deployment of a dual-teacher framework, RefKD equips the student model with the tools to introspectively analyze and reconcile the biases embedded within video datasets. This approach ensures a harmonious balance that enhances the model's retrieval capabilities across a spectrum of data distributions, from in-distribution (ID) to out-of-distribution (OOD) scenarios. Our comprehensive evaluations underscore the success of RefKD, showcasing its capability to not only maintain ID performance but also significantly elevate the model's resilience and adaptability in OOD contexts. The promising results achieved by RefKD lay the groundwork for future explorations aimed at refining the framework's efficacy and broadening its applicability. Moreover, the principles underlying RefKD hold the potential for application across various domains within video understanding, promising to address similar challenges of bias and generalization.

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

## A    APPENDIX

**Qualitative Results.** We provide the qualitative results from Charades-CD dataset in Figure 4, which verifies the robustness and effectiveness of our proposed method.

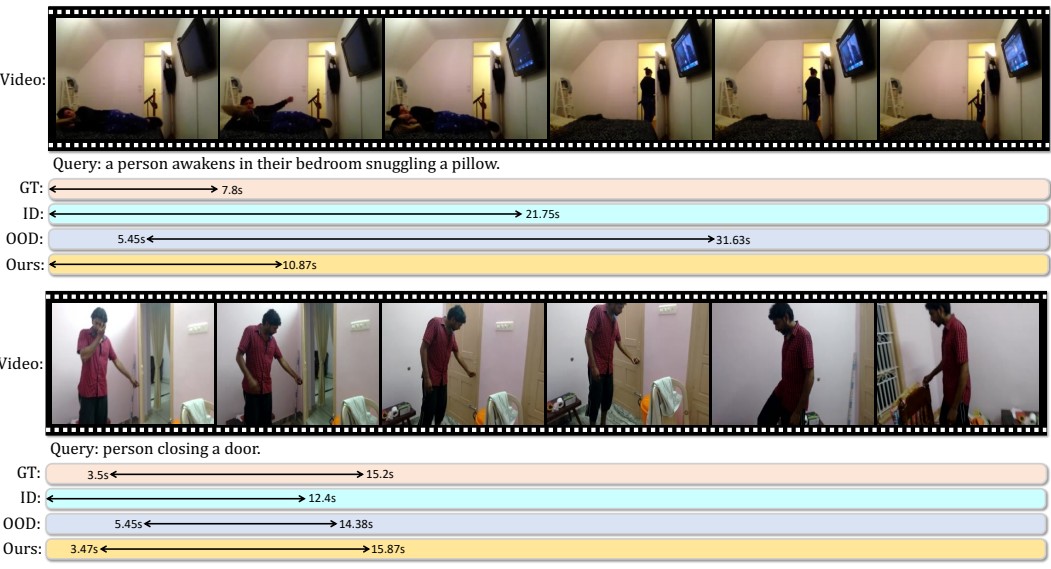

Figure 4: Visualization of Our Reflective Knowledge Distillation (RefKD) Approach. This illustration highlights the robustness and precision of RefKD in comparison to models trained solely under In-Distribution (ID) and Out-of-Distribution (OOD) conditions. The RefKD method demonstrates superior alignment with Ground Truth (GT), showcasing its enhanced adaptability and accuracy in video moment retrieval tasks.

---

**Algorithm 1:** Training Pipeline of our RefKD method.

---

**Input:** Training data $\mathcal{D}$; ID-teacher model $\mathbf{T}^I$, OOD-teacher model $\mathbf{T}^O$, and student model $\mathbf{S}$; ground-truth label $I_{s/e}$.

1   Warmup the model $\mathbf{S}$ using introspected knowledge.
2   **for** *n=1:num_epoch* **do**
3     $w_{s/e}^{ID}, w_{s/e}^{OOD} \leftarrow \text{IntroD}(\mathcal{D}, \mathbf{T}^I, \mathbf{T}^O)$;
4     **for** *d in $\mathcal{D}$* **do**
5       $\pi^{ID}, \rho^{ID} \leftarrow$ ID side-curriculum;
6       $\pi^{OOD}, \rho^{OOD} \leftarrow$ OOD side-curriculum;
7       **for** *k in $\{ID, OOD\}$* **do**
8         $\widetilde{\mathbf{P}}_{s/e}^k \leftarrow \rho^k \cdot \mathbf{P}_{s/e}^k + (1 - \rho^k) \cdot I_{s/e}$ ;
9         $\widetilde{w}_{s/e}^k \leftarrow \lambda \cdot w_{s/e}^k \cdot (1 - \frac{\pi^k}{\pi^{ID} + \pi^{OOD}})$ ;
10       **end**
11       $\widetilde{\mathbf{P}}_{s/e}^T \leftarrow \widetilde{w}_{s/e}^{ID} \cdot \mathbf{P}_{s/e}^{ID} + \widetilde{w}_{s/e}^{OOD} \cdot \mathbf{P}_{s/e}^{OOD}$ ;
12     **end**
13     $\mathcal{L} \leftarrow KL(\widetilde{\mathbf{P}}_{s/e}^T || \mathbf{P}_{s/e}^S)$ ;
14     Train model $\mathbf{S}$ using loss $\mathcal{L}$ ;
15   **end**
16   **return** Student model $\mathbf{S}$;

---

