# OpenReview forum: "Robust Video Moment Retrieval with Introspective Knowledge Distillation"
_ICLR.cc/2025/Conference — ICLR 2025 Conference Withdrawn Submission_

### Official Review · Reviewer_SwBs · 2024-11-01

**Soundness:** 2
**Presentation:** 2
**Contribution:** 3
**Rating:** 5
**Confidence:** 3

**Summary:**

The paper tackles the task of video moment retrieval where given a natural language query, the goal is to localize and retrieve specific moments from a video. It introduces RefKD that uses a dual teacher framework that distills knowledge from an In-Distribution and Out-of-distribution teacher focusing on both start and stop timestamps. Finally, the authors test the performance of the paper on two datasets.

**Strengths:**

The paper tackles an important problem and achieves good results. Overall the idea seems interesting and novel but I think some parts of the paper need to be written clearly and with less overstatemnts.

**Weaknesses:**

I find the writing style a bit forced with a lot of overstatements that I think need to be toned down in a research paper. I think a lot of parts of the paper need to be toned down in terms of language.

I also think that the data splits and how they differ from prior works needs to be better explained, since stating that the dataset was divided is not enough.

**Questions:**

1. Is there a difference between video moment retrieval and moment detection? In any way, having a short description of the differences in the related work part with some references would be good to include*.
2. Are the dataset splits made in this work or do they follow prior work? My understanding is that yes, it's split in this work to take into account OOD data. Also, how do you report numbers for the prior works if the splits are different? How are the numbers adjusted?
3. Did you test with a truly OOD teacher?

* QVHighlights: Detecting Moments and Highlights in Videos via Natural Language Queries, NeurIPS 2021
* Umt: Unified multi-modal transformers for joint video moment retrieval and highlight detection CVPR 22
* Moment Detection in Long Tutorial Videos ICCV 23

---

### Official Review · Reviewer_iGWY · 2024-11-03

**Soundness:** 2
**Presentation:** 2
**Contribution:** 2
**Rating:** 5
**Confidence:** 5

**Summary:**

In order to leverage in-distribution (ID) data for learning and maintain robustness against out-of-distribution (OOD) variations, this paper introduces a paradigm shift in VMR debiasing through Reflective Knowledge Distillation (RefKD). Based on a dual-teacher framework that includes an ID teacher and an OOD teacher, RefKD emphasizes the fusion of knowledge from both teachers, termed “INTROSPECTIVE ENSEMBLE.” Additionally, RefKD highlights the importance of rectifying the teachers’ knowledge based on the student’s confidence level regarding each teacher, referred to as “EXTROSPECTIVE ADJUSTMENT.” However, comparative experiments with the latest methods are somewhat lacking, and the manuscript suffers from imprecise expressions.

**Strengths:**

S1. This paper proposes an effective method for VMR debiasing through Reflective Knowledge Distillation (RefKD), which aligns with the natural learning process of humans.
S2. Furthermore, this paper provides an in-depth analysis of RefKD, addressing both the rationale and limitations of label rectification.

**Weaknesses:**

S1. The methods compared in Table 1 are limited to those published until 2022, resulting in a lack of comparative experiments with the latest advancements. This diminishes the relevance of the proposed method in the current research landscape.
S2. The manuscript contains several inaccuracies in its expressions. For instance, formula (4) may be incorrect, and the mention of “IntroD” in Table 4 lacks an accompanying ablation study. Such issues require careful revision to enhance the accuracy of the results.
S4. The paper employs some symbols without adequate definitions, which may confuse readers. Specifically, the meaning of “P” in sections 3.2 and 3.3 should be clarified to improve overall readability. Enhancing the clarity of these expressions would significantly benefit the audience.

**Questions:**

Q1. Could you include a comparison with more recent methods to provide a clearer context for how the proposed method performs relative to the latest advancements? This would help reinforce the method's relevance in the current research landscape.
Q2. There are some ambiguous reference in the writing, affecting the readability of this article. Please verify your writing accuracy again. For instance, the term “IntroD” in Table 4’s description appears to equate to “w/o both,” yet it lacks further explanation. Please confirm the accuracy of these references and clarify any ambiguous terms.
Q3.  Would it be possible to provide more detailed explanations or references to support the formulas and results presented, especially where they might differ from standard conventions?

---

### Official Review · Reviewer_4FhY · 2024-11-03

**Soundness:** 3
**Presentation:** 2
**Contribution:** 2
**Rating:** 3
**Confidence:** 4

**Summary:**

The paper proposes a knowledge distillation (KD) method to learn a VMR model with label rectification and introspective ensemble. This KD approach employs two teachers to distill the knowledge to a student; one for in-distribution knowledge and the other for out-of-distribution knowledge. With two modified benchmarks, Charades-CD and ActivityNet-CD, from Charades and ActivityNet, the proposed method outperforms prior arts.

**Strengths:**

* Trying to address dataset-bias for better generalization
* A decent combination of methods (two teachers, label rectification, ensemble) have been proposed

**Weaknesses:**

* Although the attempt to address dataset-bias is good, there is no clear evidence that 1) the proposed method addresses dataset-bias and 2) the reduced bias leads to better performance.
  * In L053, it seems that the method tries to address "temporal location-aware bias". But this bias is neither mentioned again nor addressed.
* Performance gain is marginal considering the complexity of the proposed method
  * How much the computational cost increases for the better performance
* There is not much of details about Charades-CD and ActivityNet-CD benchmarks. The reviewer is not very clear if the new benchmark is well defined to investigate the benefit of the proposed method.
* In L090, the authors "introduce a paradigm shift". Which paradigm are you mentioning? What is shifted? It seems just a new method of using two teacher based KD for ID/OOD method.
* Ablation shows that the proposed components do not improve the performance much. Especially, there is no evidence to address bias mitigation. (Table 3 and 4)

**Questions:**

Please address my comments in weakness section.

There are other clarification questions.
* In Fig. 2, the student model is "Span-Based". What does it mean by span-based? Why should it be?
* There is no clear description of capital bold P and capital P (without bold).
* In Eq. (1) and (2), you said the norm operator is a distance function. Is it norm operator? L1 or L2?
* In Eq. (5), $P^T$, why is it T? Is it for total?
* In Eq. (6), $P^S$, what is S?
* In L241, what is "side curriculum"? Why is it necessary? There seems no empirical study for it.
* In Eq. (8)-(10), you can write down the equation in a single row. Note that it is not a double column format.

---

### Official Review · Reviewer_xBt7 · 2024-11-04

**Soundness:** 2
**Presentation:** 3
**Contribution:** 2
**Rating:** 3
**Confidence:** 4

**Summary:**

This paper proposed a new training method called Reflective Knowledge Distillation (RefKD) for Video Moment Retrieval (VMR). The author mainly aims to address the dependency of models on training data biases and improve their generalization capabilities in practical applications. The proposed framework employed a dual-teacher structure to integrate information from different perspectives, enabling the student model to adapt to Out-Of-Distribution (OOD) data while maintaining its understanding of In-Distribution (ID) data. Experimental results on Charades-CD and ActivitityNet-CD demonstrate the effectiveness in improving both ID and OOD performance compared to existing VMR methods.

**Strengths:**

1. This paper introduced a novel training method for VMR that improves generalization capabilities, which utilizes a dual-teacher framework to integrate information from different perspectives and optimizes the student model's learning process through knowledge regularization and dynamic weighting mechanisms.

2. Detailed analysis on both ID and OOD scenarios to verify the effectiveness of the proposed method.

**Weaknesses:**

1. My biggest concern is the generalizability of the dual-teacher structure, the manner of correcting student model with ID and OOD model doesn't solve the root cause, which is the bias of the dataset. On the evaluated datasets, the effect is barely noticeable.

2. The ablation studies conducted on different backbones are limited, and the effectiveness of the method might vary depending on the specific backbone used. Additionally, this paper doesn't discuss the computational complexity of the proposed method, which could be a concern for large-scale applications.

**Questions:**

1. My biggest concern is the generalizability of the dual-teacher structure, the manner of correcting student model with ID and OOD model doesn't solve the root cause, which is the bias of the dataset. On the evaluated datasets, the effect is barely noticeable.

2. The ablation studies conducted on different backbones are limited, and the effectiveness of the method might vary depending on the specific backbone used. Additionally, this paper doesn't discuss the computational complexity of the proposed method, which could be a concern for large-scale applications.

---

### Note · Authors · 2024-11-13

I have read and agree with the venue's withdrawal policy on behalf of myself and my co-authors.